# Longitudinal dynamics of gut bacteriome, mycobiome and virome after fecal microbiota transplantation in graft-versus-host disease

Fen Zhang[1,2,5], Tao Zuo [1,2,5], Yun Kit Yeoh[1,3,5], Frankie W. T. Cheng[4], Qin Liu[2], Whitney Tang[2], Kitty C. Y. Cheung[2], Keli Yang[2], Chun Pan Cheung[2], Chow Chung Mo[4], Mamie Hui [1,3], Francis K. L. Chan[1,2], Chi-Kong Li [4], Paul K. S. Chan [1,3,6✉] & Siew C. Ng [1,2,6✉]

Fecal microbiota transplant (FMT) has emerged as a potential treatment for severe colitis associated with graft-versus-host disease (GvHD) following hematopoietic stem cell transplant. Bacterial engraftment from FMT donor to recipient has been reported, however the fate of fungi and viruses after FMT remains unclear. Here we report longitudinal dynamics of the gut bacteriome, mycobiome and virome in a teenager with GvHD after receiving four doses of FMT at weekly interval. After serial FMTs, the gut bacteriome, mycobiome and virome of the patient differ from compositions before FMT with variable temporal dynamics. Diversity of the gut bacterial community increases after each FMT. Gut fungal community initially shows expansion of several species followed by a decrease in diversity after multiple FMTs. In contrast, gut virome community varies substantially over time with a stable rise in diversity. The bacterium, *Corynebacterium jeikeium*, and Torque teno viruses, decrease after FMTs in parallel with an increase in the relative abundance of *Caudovirales* bacteriophages. Collectively, FMT may simultaneously impact on the various components of the gut microbiome with distinct effects.

[1] Center for Gut Microbiota Research, The Chinese University of Hong Kong, Hong Kong, China. [2] Department of Medicine and Therapeutics, Institute of Digestive Disease, State Key Laboratory of Digestive Diseases, LKS Institute of Health Sciences, The Chinese University of Hong Kong, Hong Kong, China. [3] Department of Microbiology, The Chinese University of Hong Kong, Hong Kong, China. [4] Department of Pediatrics, The Chinese University of Hong Kong, Hong Kong, China. [5] These authors contributed equally: Fen Zhang, Tao Zuo, Yun Kit Yeoh. [6] These authors jointly supervised this work: Paul K. S. Chan, Siew C. Ng. ✉email: paulkschan@cuhk.edu.hk; siewchienng@cuhk.edu.hk

Allogenic hematopoietic stem cell transplant (allo-HSCT) is a promising therapy for hematological disorders, however, 35–45% of recipients experience serious complications arising from allo-HSCT. These complications, termed graft-versus-host disease (GvHD), is associated with up to 25% mortality[1–3]. Emerging evidence suggests that changes in the gut microbiota composition are associated with the occurrence of GvHD[1]. Multiple factors contributed to reduced microbiota diversity in patients with allo-HSCT, including recurrent use of broad-spectrum antibiotics, administration of chemotherapy and/or radiation, and altered nutritional status[1]. In murine and human studies, allogeneic bone marrow transplantation (BMT) led to a loss of overall bacterial diversity, expansion of *Lactobacillales* and loss of *Clostridiales*[4,5]. Alterations in the gut viral community are also associated with enteric GVHD[6,7]. For instance, human herpesvirus 6 was detected in blood[6,7], and picornaviruses were detected in stools of patients with acute GvHD post HSCT[8].

Fecal microbiota transplantation (FMT), the administration of fecal suspensions from healthy donors into a patient's gastrointestinal tract, is an effective bacteriotherapy in recurrent *Clostridioides difficile* infections[9]. FMT has been shown to be associated with restoration of diversity of gut bacterial community. Recently, small case series of up to four subjects have used FMT as a treatment for refractory GvHD[10,11]. FMT was associated with improved gastrointestinal symptoms, reduced diarrhea and alterations in the gut microbiota in patients with refractory enteric GvHD[3,12].

Whilst larger clinical studies are required to confirm the safety and efficacy of FMT for GvHD, this life-threatening disease requires immediate attention in some patients. Most studies have focused on restoration of bacterial diversity and little is known of the role of gut fungi and viruses in FMT. Two studies recently reported that *Candida albicans* and bacteriophages influenced FMT efficacy in recurrent *Clostridium difficile* infections, suggesting that alterations in gut viruses and fungi in response to FMT are also important for treatment outcomes[13,14].

Here, we investigate the longitudinal dynamics of the gut bacteriome (bacterial microbiome), mycobiome (fungal microbiome) and virome (viral microbiome) in a teenager with stage IV GvHD with four doses of FMT at weekly interval. Apart from 16 S rRNA gene and shotgun metagenomics sequencing for bacteriome profiling, we enhance fungal extraction and enrich virus like particles (VLPs) in fecal DNA preparations and perform deep metagenomic sequencing on the fungi-enriched and viruses-enriched DNA extractions respectively, to profile the gut mycobiome and virome. We show discrepant alteration patterns of the gut bacteriome, mycobiome and virome in the recipient after FMT.

## Results

### Clinical characteristics of patient with severe GvHD receiving FMT.
A 14-year-old male suffering from myelodysplastic syndrome with monosomy 7 underwent human leukocyte antigen (HLA)-identical sibling allo-HSCT. The patient developed stage II skin GvHD (rash 25–50%), stage IV gut GvHD with profuse bloody diarrhea and significant functional impairment and overall grade IV life threatening GvHD shortly after allo-HSCT. The patient failed to respond to methylprednisolone, Antithymocyte globulin (ATG), cyclosporine A, infliximab, ruxolitinib and entocort (Supplementary Fig. 1a). Computed tomography imaging showed significant thickening of the small bowel and colon suggestive of severe intestinal inflammation (Supplementary Fig. 2). He received the first FMT (day 0), 77 days after allo-HSCT, followed by three additional FMTs at weekly interval

after the first FMT (Fig. 1a). The first three FMTs were performed using stool from a single donor (D8) and the fourth FMT from a different donor (D4). The antibiotic history of the patient is shown in Supplementary Fig. 1b.

### Increased diversity of gut bacterial microbiota after FMT.
We assessed alterations in the gut bacterial community in the patient before and after FMTs using 16 S ribosomal RNA gene sequencing. Changes in alpha diversity and community composition over time are summarized in Fig. 2a–c. Before FMT, only two exact sequence variants (ESVs) identified as *Corynebacterium* were detected in the patient's stools. This observation corresponded with alpha diversity metrics showing low species richness and diversity relative to donor and post FMT samples (Fig. 2a). On day one immediately after the first FMT, four *Bacteroides* and nine *Enterococcus* ESVs previously undetected in the patient's stool became dominant and *Corynebacterium* was no longer detectable (Fig. 2c). Interestingly, most of these 13 ESVs were either absent or below detection in the corresponding donor (D8) stool. The *Bacteroides* ESVs persisted in the patient's stools throughout the five days, whereas the *Enterococcus* ESVs generally declined to undetectable levels, most around day 3. Similarly, two other *Bacteroides* and one *Bifidobacterium* ESV detected in the donor stool (D8 lots 26 and 27, Fig. 2c) steadily declined in the patient from day 1 onwards. This loss of ESVs was reflected in the community alpha diversity in which richness and diversity declined after day 2 (Fig. 2a). These findings suggest that the gut microbiota of the patient may be altered as early as one day after FMT, and that not all donor microorganisms could establish colonization in the recipient.

After the second FMT using different stool from the same donor (D8), the richness and diversity of the patient's gut bacteriome increased (Fig. 2a). This time, a larger variety of Firmicutes, Fusobacteria and Proteobacteria ESVs were detected in the patient immediately after the second transplant (day 7), and most persisted until the end of the study (Fig. 2c). These observations suggest that the first FMT may have a priming effect on the patient's gut community, allowing donor microorganisms from the second FMT to colonize the recipient. Two additional FMTs were subsequently given with stool from a second donor (D4) to increase diversity of microorganisms. After the third FMT, there were no major alterations in the alpha diversity or the composition of the patient's gut bacteriome, compared to that after the first two transplants (Fig. 2a, c). A number of *Bacteroides* and *Faecalibacterium* ESVs from the fourth donor stool (donor D4) became established in the patient's gut (Fig. 2c). Overall changes of the patient's gut bacteriome were depicted via principal coordinate analysis (PCoA). The first two FMTs was associated with large shifts as shown by clustering of day 1–5 and day 7–13 profiles to the exclusion of samples from subsequent days (Fig. 2b), which was also reflected by the alpha diversity values in which community richness and diversity of the patient's stool stabilized after the second FMT (Fig. 2a). Stool bacterial communities from day 67 onward appeared to cluster away from samples immediately following the fourth FMT, suggestive of fluctuations in the patient's gut microbial communities. Nevertheless, these post-FMT mcirobiome profiles did not resemble disease baseline microbiome profile (day 0 before FMT).

To gain high-resolution insight into the bacterial microbiota, a subset of samples were analyzed by metagenomic sequencing and the results showed similar alterations with that measured via 16 S rRNA sequencing after FMT (Supplementary Fig. 3). Three predominant species were detected in the recipient's stool after FMTs, including *Alistipes putredinis*, *Clostridium nexile* and *Ruminococcus gnavu* (Supplementary Fig. 3).

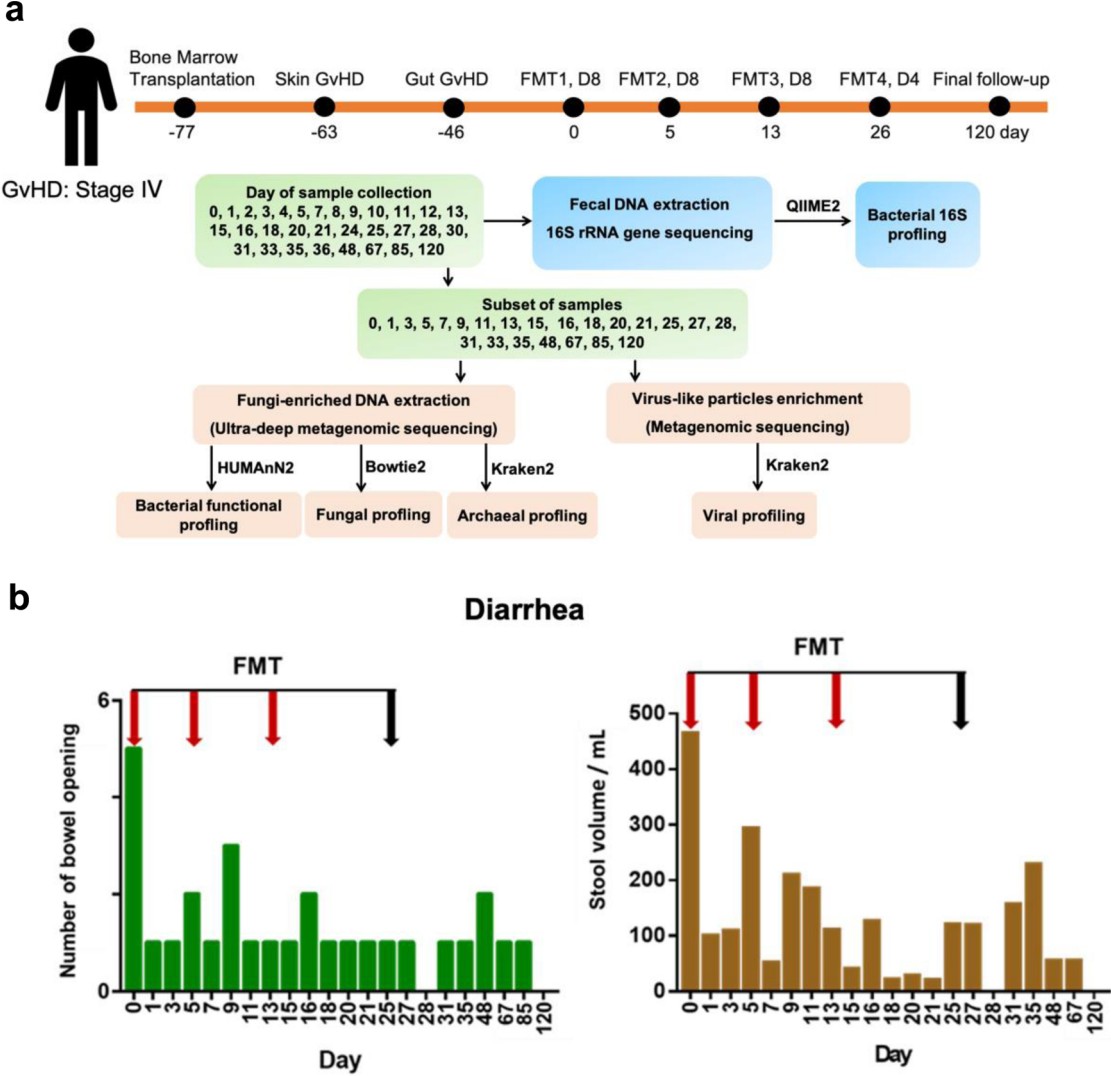

**Fig. 1 Study schematic and clinical outcome. a** GvHD patient received four FMTs. The first three (FMT1, FMT2, FMT3) were performed using stool from a single donor, D8 and the fourth (FMT4) from a different donor D4. **a** Timeline of HSCT, GvHD onset, FMT treatment, stool collection and sequencing strategy. **b** Number of bowel openings and stool volume of the patient before and after FMT. Red arrows represent fecal transplant using donor D8, black arrow represents fecal transplant using donor D4. One biological sample was examined over one experiment.

**Improved gut microbiome functionality after FMT**. We next investigated functional alterations in the bacterial community following FMTs, total bulk DNA sequencing reads from the metagenomic dataset were queried against reference databases to identify genes and their corresponding metabolic pathways (MetaCyc annotation). In the stool collected at baseline (day 0 before FMT), a large proportion of sequences could not be classified (>95% unmapped, Supplementary Fig. 4). The richness of genes and pathways showed no great difference between donors and patient before and after FMT (Supplementary Fig. 5). However, Shannon and Simpson diversity of the genes and pathways in the patient were markedly lower than that in donor (Fig. 3a), which is consistent with a low taxonomic diversity (Fig. 2a). After the first FMT, there was a substantial increase in the proportion of classifiable genes and pathways (Supplementary Fig. 4). In contrast to taxonomic reconstitution in which only several ESVs from the genera *Bacteroides* and *Enterococcus* were detected in the patient (Fig. 2c), Shannon and Simpson diversity of genes and pathways were increased after the first FMT (Fig. 3a), and the overall functional gene composition was comparable to the donors (Fig. 3b). After the second FMT, the

diversity of genes and pathway remained stable until the last follow-up, accompanied by an increase in the bacterial diversity at the taxonomical level. After FMT, genes and pathways primarily related to metabolisms of lipids, carbohydrates, amino acids, nucleotides and energy were restored (Fig. 3c). These findings suggest that functions of microbial community might be promptly reconstructed following FMT with less fluctuations in functional content when compared with taxonomic configuration.

**Durable engraftment of donor-derived fungi after FMT**. To characterize the changes of the gut mycobiota in the patient following multiple FMTs, we performed fungi-enriched fecal DNA extraction, ultra-deep metagenomic sequencing and profiled the fungal composition. Overall, the patient showed a decrease in the richness (Chao1) and diversity (both Shannon and Simpson indices) of the gut mycobiome after FMT (Fig. 4a). The fecal mycobiome compositions at later time points after serial FMTs were comparable to the fecal mycobiomes of the donor D8 (D8-26 and D8-34) (Fig. 4b).

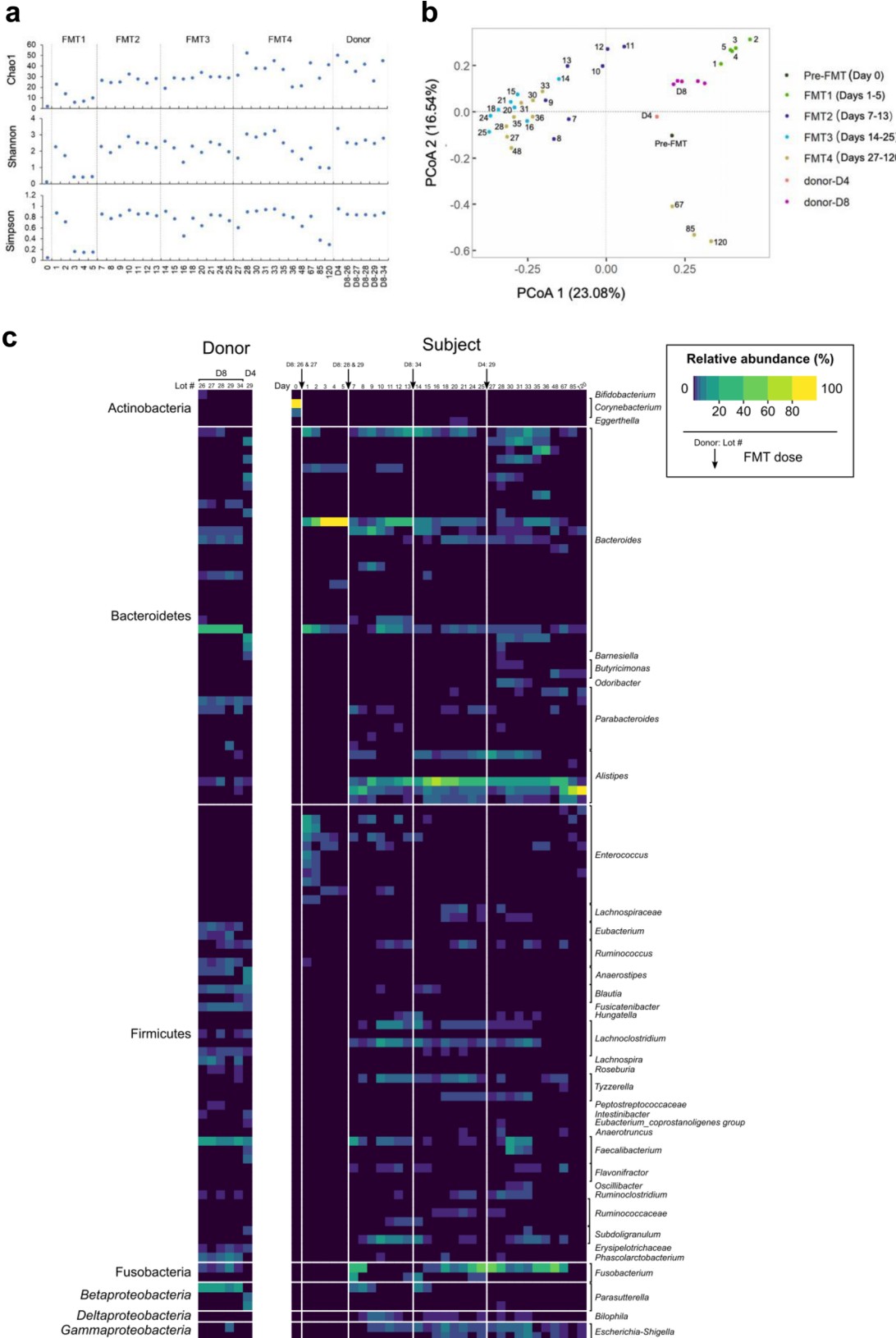

We further investigated the proportion of recipient fungal community that was derived from donors by fast expectation-maximization microbial source tracking (FEAST). Donor derived fungi instantly took up a proportion as high as 71% in the recipient's mycobiome on day 1 after FMT (Fig. 4c), followed by a decrease down to 10.3% on day 5, indicating that most of fungal taxa showed transient colonization in the recipient. With subsequent FMTs, the proportion of donor-derived fungal taxa continued to rise followed by a marked decline to 4% on day 120 which suggest that only limited fungal taxa from donors have sustained engraftment in the recipient.

**Fig. 2 Changes in gut bacterial community composition in GvHD patient after FMTs. a** Richness, Shannon and Simpson diversity of bacterial communities in donor stools and the patient before and after FMTs. **b** Principal coordinate analysis (PCOA) of bacterial community composition in the patient following FMT based on Bray-Curtis dissimilarities. **c** Heat map summarizing fecal bacterial community composition in donor stools and the patient. Increasing relative abundances are representing by lighter colors (black to yellow). Taxonomic labels on left of figure indicate phylum assignments (class-level for Proteobacteria), labels on right indicate genus. Pre-FMT represents stool sample collected before the first FMT. FMT1, FMT2, FMT3, and FMT4 represent stool samples collected after the 1st (days 1–5), 2nd (day 6–13), 3rd (day 14–25) and 4th (day 27–120) FMTs respectively. D8-26, D8-27, D8-28, D8-29, D8-34 indicate fecal samples collected from donor D8 on different days, whereas only one fecal sample collected from donor D4 was used in the FMT. FMT1, FMT2, FMT3, FMT4 were performed using D8-26 and D8-27, D8-28 and D8-29, D8-34, D4 respectively. All analyses were based on bacterial profiles from 16 S rRNA gene sequencing data.

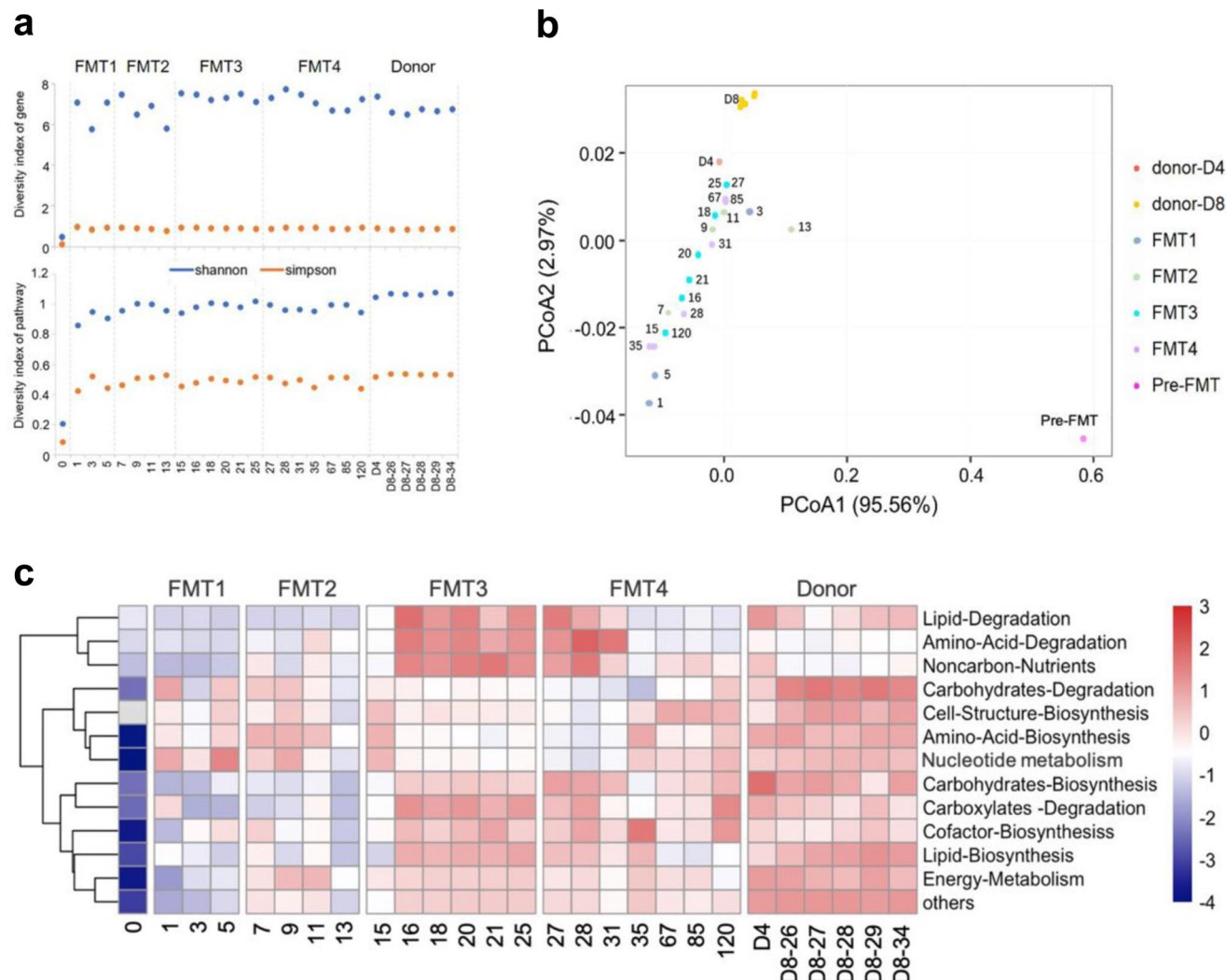

**Fig. 3 Changes in bacterial community function in GvHD patient associated with FMTs. a** α-diversity (Shannon and Simpson diversity) plot of gene families and pathways. These values were calculated based on gene and pathway relative abundances. **b** Principal coordinate analysis of Bray-Curtis dissimilarities based on abundances of functional pathways detected in donor and patient stools. Pre-FMT represents stool sample collected before the first FMT. FMT1, FMT2, FMT3, and FMT4 represent stool samples collected after the 1st (day 1-5), 2nd (day 7-13), 3rd (day 15-25) and 4th (day 27-120) FMTs respectively. **c** Heat map summarizing changes in patient's gut microbial community function following FMT. Pathways with higher relative abundances are shaded red, whereas pathways with low relative abundances are shaded blue.

At the taxonomy level, the species *Parastagonospora nodorum* and *Thielavia terrestris* significantly dominated the patient's fungal community before FMT but were deceased following FMT (Fig. 4d). *Saccharomyces cerevisiae*, which dominated the donors' fungal community, expanded rapidly on day 1 after the first FMT and decreased after the third FMT in the patient. In contrast, *Candida dubliniensis* was increased after FMT and remained stable until the last follow up. After the third FMT, *Sporisorium reilianum* derived from the donor showed high abundance in the

patient's gut after the third FMT but was depleted after the fourth FMT (on day 35 since the first FMT). This was concomitant with a post-FMT recovery of *Sclerotinia sclerotiorum*, which showed a high relative abundance before FMT. We further performed quantitative PCR on a subset of these significantly altered species, *Parastagonospora nodorum*, *Thielavia terrestris*, *Candida dubliniensis* and *Saccharomyces cerevisiae*. Consistent with the sequencing results, *Parastagonospora nodorum* and *Thielavia terrestris* were significantly decreased, *Candida dubliniensis* was

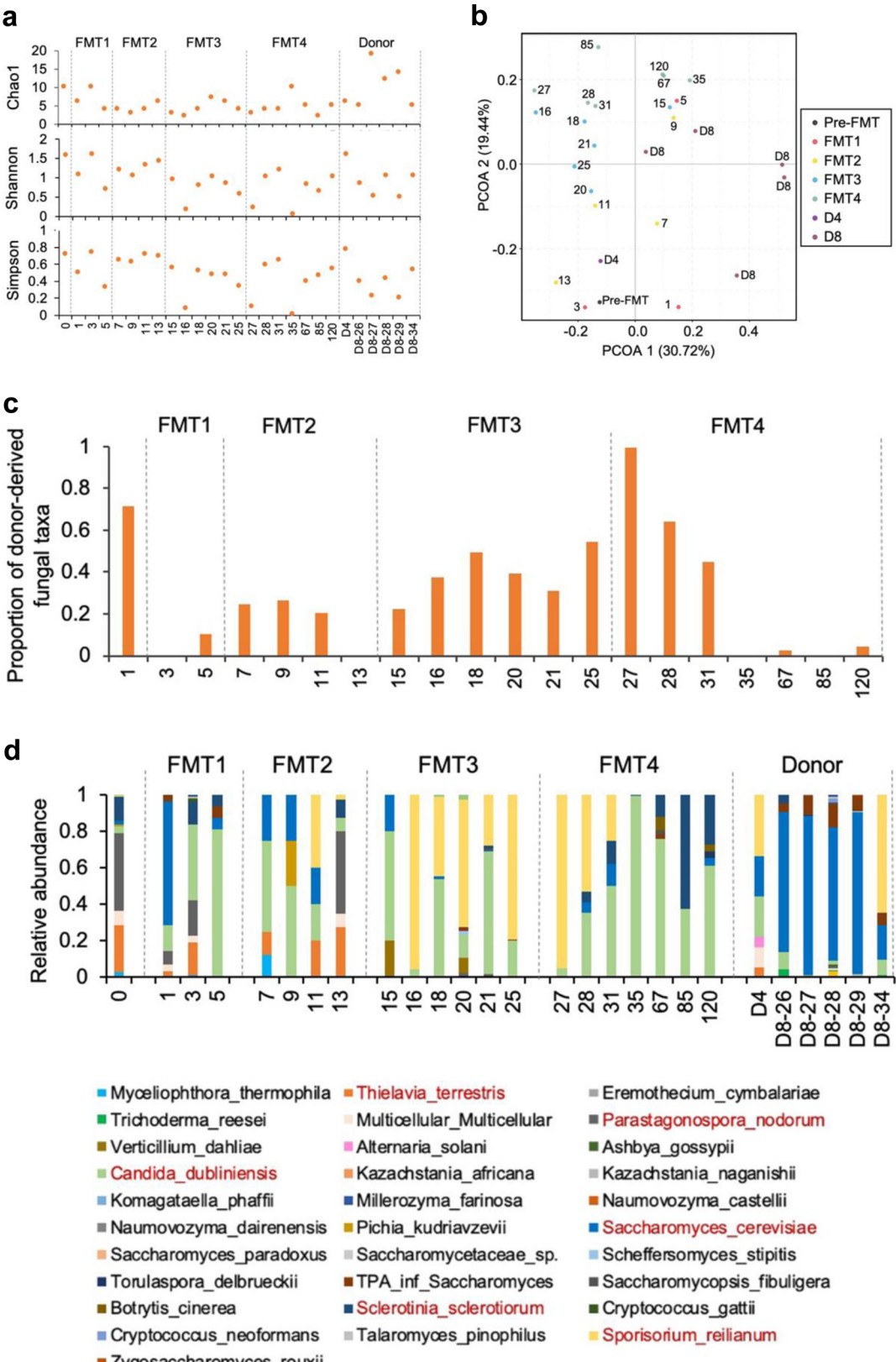

**Fig. 4 Changes in gut fungal community composition in GvHD patient after FMTs. a** Richness, Shannon and Simpson diversity of fungal communities in donor stools and the patient before and after FMTs. **b** PCoA ordination based on Bray-Curtis dissimilarities of the patient's gut mycobiota showing progression over time. Dots represent samples labeled according to their respective time-points and are colored to visually group samples according to FMT timeline. **c** Proportion of fungal taxa derived from donors in total taxa in patient's stool samples. The proportion was calculated by using the method of fast expectation-maximization microbial source tracking (**FEAST**). **d** Species-level proportions of the fecal mycobiota in donors and the patient before and after FMT. The markedly altered species were highlighted in red color. One biological sample was examined over one experiment.

increased, whereas *Saccharomyces cerevisiae* was transiently expanded in the feces of the patient after FMT (Supplementary Fig. 6). These findings suggest that FMT may be associated with an expansion of multiple fungal taxa, rather than a recovery of fungal diversity.

To exclude possibility of fungal contamination, we sequenced negative control samples during DNA extraction. The most abundance species in negative controls, such as *Talaromyces pinophilus*, *Cryptococcum neoformans* and *Alternaria solani*, were not detected in any patient's or in donors' stool samples (Supplementary Fig. 7). Moreover, the significantly altered species, *Parastagonospora nodorum*, *Thielavia terrestris*, *Candida dubliniensis* and *Saccharomyces cerevisiae* were not detected in the negative control samples by quantitative PCR (Supplementary Fig. 6). These results suggest that the fungal alteration findings reported here are unlikely to be the result of contamination.

**Variable and stable increase in virome diversity after FMT.** To characterize alteration in the gut virome, we enriched virus like particles (VLPs) in stool samples and subsequently performed deep metagenomic sequencing on the VLP DNA. The patient's virome exhibited a high degree of variability in virome diversity over time (Supplementary Fig. 8a) and the overall community composition structure after FMT did not resemble that of the donors' with the exception at day 15 after the third FMT and at day 35, 48 and 67 after the fourth FMT (Supplementary Fig. 8b). We further investigated the proportion of viruses that was derived from donors in the recipient after FMT. Donor's viruses gradually took up greater proportions in recipient after each FMT (Fig. 5a). Less than 2% of the recipient's viral taxa was transferred from donor after the first and second FMTs, whereas after the third FMT, the proportion of donor-derived viral taxa was increased up to 29% by months 3.

At day 0, the virome predominantly was comprised of eukaryotic viruses (99.99% of total mapped reads) and few phage reads were detected (0.01%). After the first and second FMT, more than 99% of virus sequences detected were still eukaryotic viruses. Most of the reads were identified as *Torque teno* virus (Fig. 5c, relative abundance 99.67% in patient versus 0.0834% in donors). This eukaryotic virus, *Torque teno* virus, maintained high relative abundance in the gut virome after the first and second FMT but was substantially reduced after the third and fourth FMT to an average relative abundance of 53.72% over the period of day 27 to 120. The abundance of *Torque teno* virus in the feces of patients and donors were further quantified by qPCR, which showed a 2000-fold higher presence in the patient before FMT than that in donors. However, it was decreased shortly after each FMT, and finally maintained at lower levels after 31 day post the first FMT (average 150-fold decrease over the period of day 31 to day 120), compared to the baseline level in the patient before FMT (Supplementary Fig. 9). We reported alterations of the most dominant 50 eukaryotic viral species following FMT, and found that eukaryotic viruses other than *Torque teno* virus were presented in low abundances and did not display significant changes up to three months after FMT (Fig. 5f).

The relative abundances of phages (prokaryotic viruses) elevated since the third FMT and reached to 41.82% on day 120 after the first FMT (Fig. 5b). Along with slow increases in the relative abundance of phages over the treatment period, a substantial increase in phage diversity was also observed in the patient's gut virome after the first FMT (Fig. 5d). Members of the bacteriophage lineage *Microviridae* occupied a large proportion of the patient's prokaryotic virome before FMT (88.13% in relative abundance), but was subsequently replaced (<10% average relative abundance after FMT) by *Caudovirales* (11.86% in

relative abundance before FMT, >90% in final sample) (Fig. 5e). At the species level, the patient's prokaryotic viral community was dominated by *Parabacteroides phage YZ-2015a* (74.3%) and *Parabacteroides phage YZ-2015a* (13.9%), which both decreased to 0% on day 120 after FMT. After the second FMT, five *Enterobacteria phage* taxa, one *Escherichia phage*, two *Stx2 converting phage* and one *Shigella virus* were expanded transiently, but were depleted after the forth FMT. In contrast, the five *Enterococcus phage* taxa maintained at a high relative abundance from day 67 to 120 (Fig. 5g). To exclude the possibility of contamination, we sequenced a negative control sample (reagent control) during VLP extraction and amplification. Only five species were detected in the negative control, which collectively took up <0.4% of the virome across donors and recipient samples (Supplementary Fig. 10). These data suggest that viral alterations reported here are unlikely to be the result of contamination.

## Discussion

Resident members of the gut microbiota including bacteria, fungi and viruses are intimately linked with host immunity and human health[15,16]. The basis for FMT is to alter these microorganisms and their interactions by replacing a "dysbiotic" gut microbiota with "healthy" microbes to treat gut and other non-gut diseases. GvHD is traditionally considered as an immunological disease but can be treated by FMT with promising outcomes[10,11]. Previous reports have focused on the bacterial community after FMT in GvHD[10,11], whereas the role of gut fungi, viruses in GvHD remain unknown.

Our previous study reported that *Candida albicans* and bacteriophages can impact FMT efficacy in recurrent *Clostridium difficile* infections[13,14]. Here, we investigated longitudinal dynamics of the gut bacterial, fungal, and viral communities in a GvHD patient treated with multiple FMTs. We found that the bacterial, fungal, and viral communities responded differently to FMT; The bacterial community in the recipient was gradually recovered with an increased diversity after each FMT, the fungal community showed expansion of several species followed by a decrease in diversity, whereas the viral community varied substantially in composition after multiple FMTs with a stable rise in virome diversity.

Initially, the GvHD patient's gut bacterial microbiota was characterized by a low diversity, and dominated by *Corynebacterium jeikeium*, which has been reported to be associated with bacteremia in catheter manipulation[17]. Given the toxin-producing capability of *Corynebacterium* which may lead to diarrhea, we performed PCR to identify diphtheria toxin encoding genes in the patient's fecal microbiome before FMT. We found absence of diphtheria toxin encoding genes, indicating that the diarrhea in the patient might not be caused by *Corynebacterium* GI infection but by GvHD (Supplementary Fig. 11).

Gut bacterial-fungal interactions is important in maintaining mucosa homeostasis[18]. In our study, the mycobiota change is opposite to the bacterial microbiota change. We were unable to determine whether the decreased mycobiota diversity is a result of reconstitution of microbial taxa, replacement of donor's fungal community, or a recovered host immune system. Nonetheless, several fungal species were increased after FMT, such as *Candida dubliniensis* and *Sporisorium reilianum*. Alterations of fungal composition associated with FMT warrant further mechanistic investigations to delineate their significance and implications in disease outcomes.

Both eukaryotic and prokaryotic DNA viruses (mostly phages) are constituents of the human gut microbiota, where eukaryotic DNA viruses are minor components compared to bacteriophages[19]. However, we found a substantial presence of

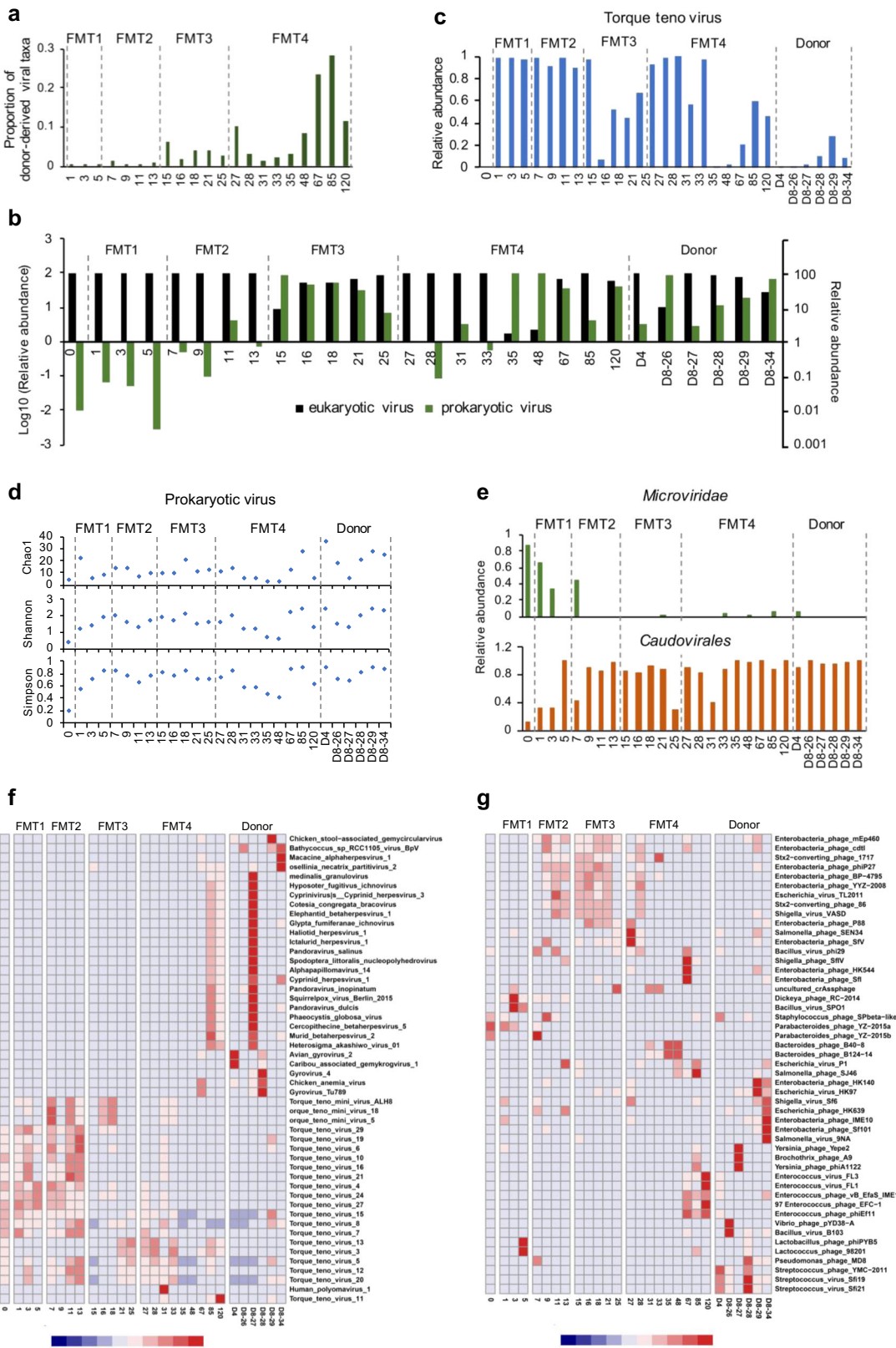

eukaryotic viruses relative to bacteriophages in the patient prior to FMT, whereby the use of immunosuppressants may be associated with the expansion of *Torque teno* virus[20]. *Torque teno* virus was previously reported to be increased in GvHD patient[21,22], but it is uncertain whether it associates with occurrence and/or cure of GvHD.

FMT was associated with an increase in the richness of viruses and the ratio of bacteriophages to eukaryotic viruses in the patient's gut virome. However, the enrichment and/or depletion of exact viral taxa associated with recovery remains to be investigated. *Caudovirales* bacteriophages were overrepresented in patients with *Clostridium difficile* infections and were significantly

**Fig. 5 Changes in gut viral communities in GvHD patient after FMTs. a** Proportion of viral taxa derived from donors in total taxa in patient's stool samples. The proportion was calculated by using the method of fast expectation-maximization microbial source tracking (FEAST). One biological sample was examined over one experiment. **b** Relative abundances of eukaryotic and prokaryotic viruses in the donor and patient fecal samples. One biological sample was examined over one experiment. **c** Changes in relative abundance of a *Torque teno virus* in the patient. One biological sample was examined over one experiment. **d** Richness, Shannon and Simpson diversity of phages in donor and patient stools. **e** Relative abundances of *Microviridae* and *Caudovirales* phages in donor and patient stools. **f** Heatmap summarizing chronological alterations of 50 most dominant eukaryotic viral species in the patient's viral community following FMTs at the species level. One biologically independent sample examined over one independent experiment. **g** Heatmap summarizing chronological alterations of 50 most dominant prokaryotic viral species in the patient's viral community following FMTs at the species level. The abundance of selected species in a given sample is color-intensified according to the row z-score value, as indicated in the color bar. Blue to red color shades indicate increasing relative abundances.

reduced by FMT[13]. Whereas here we observed a reverse trend of FMT enriching *Caudovirales* phages in the GvHD patient. As the role of viruses in FMT has not been well studied, future research should focus on changes in the viral community and their relationship with FMT efficacy. Moreover, we did not study RNA viruses which could be potentially important as well in FMT. Such information will be crucial in understanding and enhancing the safety and efficacy of FMT for the treatment of various diseases. The observational nature of this single case precludes generalization of our findings and further larger sample-sized studies are needed to validate the present findings.

In conclusion, our study provides novel insight into alterations of the gut microbiome in GvHD following multiple FMTs. We showed that bacterial, fungal and viral communities responded differently to FMTs. Future FMT practice should consider the role and importance of reconstituting the gut fungi and viruses.

## Methods

**Ethics statement**. This study was approved by Joint Chinese University of Hong Kong-New Territories East Cluster Clinical Research Ethics Committee (The Joint CUHK-NTEC CREC, CREC Ref. No.: 2017.260 and 2018.443, Clinical Trial registry, NCT04014413). The patient and his guardians consented to participate in this study and agreed for publication of research results.

**Preparation of donor stools for FMT**. Donor stools were obtained from the stool bank of Center for Gut Microbiota Research. To ensure donors were healthy and fit for stool donation, donors were first screened using a questionnaire followed by stool and blood tests to rule out any infectious disease such as HIV, hepatitis B or C, syphilis or any communicable diseases such as inflammatory bowel disease, irritable bowel syndrome, or gastrointestinal malignancy. Shortly after donors provided stool at the Prince of Wales Hospital, samples were diluted with sterile saline (0.9%). This solution was blended and strained through a filter to remove particulate matter. The resulting supernatant was then stored as frozen FMT solutions.

**Study subject and design**. A 14-year-old male suffering from myelodysplastic syndrome with monosomy 7 underwent HLA-identical sibling allo-HSCT. The patient developed itchy erythematous and maculopapular skin rash over all limbs and body covering 50% of body surface area from Day 14 post-transplant (Day -63, Fig. 1a) and at the time of neutrophil engraftment. The clinical picture was compatible with stage 2 skin GvHD according to the Glucksberg clinical stage of acute GvHD. The patient was treated with intravenous methylprednisolone 2 mg/kg/day with clinical response and the skin rash gradually subsided after steroid treatment. However, at the time of tapering of steroid to 1 mg/kg/day on Day 31 post-transplant (Day -46, Fig. 1a), the patient developed severe abdominal pain, vomiting and diarrhea. The watery diarrhea progressed to bloody diarrhea despite aggressive immunosuppressive treatment. CT scan showed abnormal mucosal enhancement and diffuse long segment inflammatory changes in the small bowel involving whole length of ileum and jejunum with intraluminal hyperdense content consistent with hemorrhage. The large bowel also showed increase in wall thickening and mucosal stratification (Supplementary Fig. 2). Sigmoidoscopy showed hemorrhage in the large colon with mucosal sloughing hence biopsies were not taken due to increased risk of perforation. All infective pathogens including clostridium difficile toxins were excluded in stool and the overall diagnosis was compatible with gastrointestinal GvHD.

The patient was administered four FMTs, and recovery was monitored through subsequent clinical follow-ups (Fig. 1a). Firstly, a pre-FMT stool sample was obtained from the patient on day 0. FMTs were then administered to the patient on day 0, day 5, day 13 and day 25 of the study. We used five stool samples (D8-26, D8-27, D8-28, D8-29, D8-34) from one donor (D8) and one stool sample from

another donor (D4). The first, second, third and fourth FMTs were performed using D8-26 and D8-27, D8-28 and D8-29, D8-34, and D4 stools, respectively. During the FMT procedure, 100 ml of FMT solution was infused over 2–3 min into the distal duodenum or jejunum of the patient via oesophago-gastroduodenoscopy (OGD). Post-FMT stools were collected daily from day 1 to day 36, and every half month after the fourth FMT until day 120. The FMT treatment in this patient was experimental. The study complied with the CONSORT guidelines of N-of-1 clinical trial, and pre-specified outcomes related to this patient as part of the broader trial (NCT04014413) will be reported upon trial completion.

**16 S rRNA gene sequencing and analysis**. The total DNA was extracted from around 0.1 g of stool using the DNeasy PowerSoil Kit (Qiagen catalog number 12888-100) following manufacturer's instructions. Extracted fecal DNA was used as template in a two-step PCR to amplify the V3 and V4 regions of the 16 S gene. Briefly, PCR reactions were performed in 25 µl reactions containing 1X PCR buffer, 1 mM MgCl$_2$, 0.2 mM of each dNTP, 0.2 µM of each forward (341 F: CCTACGGGNGGCWGCAG) (Klindworth et al., 2013) and reverse (806 R: GGACTACNVGGGTWTCTAAT) (Apprill et al., 2015) primers, 0.625 U HotStarTaq *Plus* DNA Polymerase (Qiagen catalog number 203605), 1 µl template DNA and molecular biology grade water. Thermocycler settings were as follows: 95 °C for 15 min, followed by 15 cycles of 94 °C for 1 min, 55 °C for 1 min and 72 °C for 3 min, and then 10 cycles of 94 °C for 1 min, 60 °C for 1 min and 72 °C for 3 min, and finally 72 °C for 10 min. The second PCR to ligate sequencing adapters and barcodes to amplicons from the first PCR were performed in 25 µl reactions with the following recipe: 1X PCR buffer, 1 mM MgCl$_2$, 0.2 mM of each dNTP, 0.2 mM each of Nextera index 1 and 2 primers, 0.1 µM each of forward and reverse Golay indexes, 0.625 U HotStarTaq *Plus* DNA Polymerase, 1 µl template DNA from the first PCR and molecular biology grade water. Thermocycler settings were 95 °C for 3 min, followed by seven cycles of 98 °C for 20 s, 63 °C for 30 s and 72 °C for 1 min, and finally 72 °C for 5 min. PCR amplicons were checked on an agarose gel and purified using the QIAquick Gel Extraction Kit (Qiagen catalog number 28706). Purified final products were sent to the Genomics Resource Core Facility at Cornell University for 2 × 300 bp sequencing in an Illumina MiSeq using the v3 Miseq Reagent Kit. Demultiplexed raw sequence data were imported into QIIME2 v2018.6 (https://qiime2.org/). Using the DADA2 workflow[23] in QIIME2, PCR primer and low-quality sequences were trimmed, and remaining reads subsequently denoised and merged. Alpha diversity statistics were calculated using representative sequences produced by DADA2. A taxonomy classifier trained on V3 and V4 regions of reference 16S sequence[24] (SILVA release 128) was then used to assign taxonomic identities to the representative sequences. The final site-by-species counts table based on exact sequence variants (ESVs)[25] was exported from QIIME2, chloroplast and mitochondrial counts removed, and used as input into R for statistical analysis.

**Fungi-enriched fecal DNA extraction**. Fecal DNA was extracted by using Maxwell® RSC PureFood GMO and Authentication Kit (Promega) with some modifications to increase the yield of fungi DNA. Approximately 100 mg from each stool sample was prewashed with 1 ml ddH$_2$O and pelleted by centrifugation at 13,000 × $g$ for 1 min. The pellet was resuspended in 800 µl TE buffer (pH 7.5), supplemented with 1.6 µl 2-mercaptoethanol and 500 U lyticase (Sigma) digesting cell walls of fungi, and incubated at 37 °C for 60 min, which increase the lysis efficacy of fungal cell. The sample was then centrifuged at 13,000 × $g$ for 2 min and the supernatant was discarded. After this pretreatment, DNA was subsequently extracted from the pellet using a Maxwell® RSC PureFood GMO and Authentication Kit (Promega) following manufacturer's instructions. Briefly, 1 ml of CTAB buffer was added to the pellet and vortexed for 30 s, then the solution heated at 95 °C for 5 min. After that, samples were vortexed thoroughly with beads (Biospec, 0.5 mm for fungi and 0.1 mm for bacteria,1:1) at maximum speed for 15 min. Following this, 40 µl proteinase K and 20 µl RNase A were added and the mixture Incubated at 70 °C for 10 min. The supernatant was then obtained by centrifuging at 13,000 × $g$ for 5 min and placed in a Maxwell® RSC instrument for DNA extraction. The extracted fecal DNA was used to preform metagenomics sequencing.

**Virus-like particles enrichment**. Virus-like particles (VLPs) were enriched and other DNA was eliminated according to a protocol described in a previous study[13]. A 400 μl volume of saline-magnesium buffer (0.1 M NaCl, 0.002% gelatin, 0.008 M $MgSO_4H_2O$, 0.05 M Tris pH7.5) was added to 200 mg of stool sample and vortexed for 10 min. The sample was then centrifuged at 2,000 x $g$ and the pellet was discarded. To remove bacterial cells and other residual material, the suspension was filtered through a 0.45 μm and two 0.22 μm filters. The cleared suspension was incubated separately with lysozyme (1 mg/ml at 37 °C for 30 min) and chloroform (0.2x volume at RT for 10 min) to degrade any remaining bacterial and host cell membranes. A DNase cocktail including 1U Baseline zero DNase (Epicenter) and 10U TurboDNaseI (Ambion) was added into the sample and the mixture was incubated at 65 °C for 10 min to eliminate DNA non protected within virus particles. VLPs were lysed (4% SDS plus 38 mg/ml Proteinase K at 56 °C for 20 min), treated with CTAB (2.5% CTAB plus 0.5 M NaCl at 65 °C for 10 min), and nucleic acids were extracted with phenol: chloroform pH 8.0 (Invitrogen). The aqueous fraction was washed once with an equal volume of chloroform, purified and concentrated on a column (DNA Clean & Concentrator TM 89-5, Zymo Research). VLP DNA was amplified for 2 h using the Phi29 polymerase (GenomiPhi V2 kit, GE Healthcare) prior to sequencing. Four independent amplification reactions were performed for each sample and pooled together to reduce amplification bias.

**Metagenomics sequencing and analysis**. Fungi-enriched DNA and VLP-enriched DNA were sheared into fragments of 150 bp VLP by ultrasonication (Covaris), and used to construct sequencing libraries. Libraries were prepared through the processes of end repairing, A-tailing, adapter ligation, size-select library and PCR amplification. DNA libraries from fungi-enriched DNA were sequenced using the Illumina HiSeq X Ten at the Beijing Genomics Institute (BGI) to generate 150 bp paired-end reads, yielding an average 67 ± 8.1 million reads (12 G data) per sample (Supplementary Table 2). VLP-enriched DNA libraries were sequenced (also 150 bp paired-end) by on the Illumina NovaSeq 6000 at Novogene, Beijing, China. An average of 26 ± 3.3 million reads (6 G data) per sample were obtained (Supplementary Table 3).

Raw sequence reads were filtered and quality-trimmed using *Trimmomatic v0.36*[26] as follows: 1. Trimming low quality base (quality score < 20), 2. Removing reads shorter than 50 bp, 3. Tracing and cutting off sequencing adapters. Contaminating human reads were filtering using *Kneaddata v0.7.3* (https://bitbucket.org/biobakery/kneaddata/wiki/Home, Reference database: GRCh38 p12, https://www.ncbi.nlm.nih.gov/assembly/GCF_000001405.38/) with default parameters.

Profiling of bacterial taxonomy and functional composition was extracted using *humann2 v0.11.1*[27] from metagenomes of fungi-enriched DNA, which included taxonomic identification via *MetaPhlAn2* by mapping reads to clade-specific markers[28], annotation of species pangenomes through *Bowtie 2 v2.3.5*[29] with reference to the ChocoPhlAn database (http://huttenhower.sph.harvard.edu/humann2_data/chocophlan/chocophlan.tar.gz), translated search of unmapped reads with *DIAMOND v2.0.4*[30] against the UniRef90 universal protein reference database (https://www.uniprot.org/help/uniref)[31], and pathway collection from the generated gene list with reference to the Metacyc database (https://metacyc.org/)[32]. Gene families and pathway abundances estimated from the data were normalized according to relative abundances of their corresponding microbial taxa reported by MetaPhlAn2 v2.96.1-0.

Taxonomic profiling of fungi was performed on metagenomes sequenced from fungi-enriched DNA, using *HumanMycobiomeScan*[33]. The database was based on the complete fungal genomes available at the NCBI website (downloaded in January 2019). A second database containing 38,000 entries, corresponding to 265 different fungal genomes, was available for download (referred to as Fungi_FULL) on the project web page (https://sourceforge.net/projects/hmscan/). Briefly, metagenomic reads were aligned to the fungal genome database using via *Bowtie 2 v2.3.5*[34] to identify candidate fungal reads. Afterwards, sequences were trimmed for low quality scores (less than 3) and reads shorter than 60 bases are discarded. To remove human and bacterial contaminations, a double filtering step was performed using *BMTagger v3.101-1* against the hg19 database for human sequences (https://www.ncbi.nlm.nih.gov/assembly/GCF_000001405.13/) and a custom bacterial database which used for *ViromeScan*[35]. Then, the filtered reads were matched again to the fungal database using *Bowtie 2*[34] for definitive taxonomic assignment. The taxonomic affiliation was deduced by matching the result of the taxonomic assignment with an annotated list of fungal species, containing the entire phylogenetic classification for each genome included in the database. At the end of the process, the results were normalized by the length of the references included in the database, and the relative abundance profiles were obtained.

Taxonomic profiling of virus was performed on metagenomes sequenced from VLP-enriched DNA, using *Kraken 2 v2.0.7-beta*. The complete NCBI viral RefSeq databases (accessed on October 20, 2018)[36] were downloaded (https://www.ncbi.nlm.nih.gov/refseq/). *Kraken 2* examined the k-mers within a query sequence and used the information within those k-mers to query a database. sequence reads were mapped to the lowest common ancestor of all reference genomes with exact k-mer matches. Each query was thereafter classified to a taxon with the highest total hits of k-mers matched by pruning the general taxonomic trees affiliated with mapped genomes.

**Quantitative PCR for detection of Parastagonospora nodorum, Thielavia terrestris, Candida dubliniensis and Saccharomyces cerevisiae in fungi-enriched fecal DNA**. Relative quantification was achieved using the standard curve technique.

*Parastagonospora nodorum, Thielavia terrestris, Candida dubliniensis and Saccharomyces cerevisiae* were quantified by SYBR Green I assays using the SYBR® Premix Ex Taq ™, and total fungi was quantified by TaqMan assays using Taqman Premix Ex Taq ™ (TaKaRa). Standard curves were established by qPCR of serial dilutions of pUC57 plasmids harboring target genes (Supplementary Table 4). The sequences of the primers and genes which were used to generate standard curve in this study are detailed in Supplementary Table 4. The mass of four target species and total fungi were obtained through interpolation from the standard curve using CT value. Then the mass of three target species in each sample can be normalized by their respective fungal mass to calculate a normalized target value (Normalized target = Mass of each fungal species/Mass of total fungi). The normalized target values in the fecal samples from patient after FMT or donor are divided by normalized target values in patient's baseline sample to calculate the fold change (Fold change = Normalized target of post-FMT samples or donors' sample / Normalized target pre-FMT samples).

**Quantitative PCR for detection of Torque teno virus load in human fecal Virus-like particles DNA**. Total Torque teno virus (**TTVs**) loads in patient and donor were quantified by TaqMan qPCR analysis (Premix Ex TaqTM, TaKaRa) using primers[22]: forward primer 5′-GTGCCGIAGGTGAGTTTA-3′; reverse primer 5′-AGCCCGGCCAGTCC-3′; probe 5′-TCAAGGGGCAATTCGGGCT-3′. The absolute copy numbers of TTVs were determined from standard curves established by qPCR of serial dilutions of pUC57 plasmids harboring target genes.

**PCR detection of diphtheria toxin gene in fungi-enriched DNA**. The PCR assay was performed by using Primers, Tox1(ATCCACTTTTAGTGCGA-GAACCTTCGTCA) and Tox2 (GAAAACTTTTCTTCGTACCACGGGACTAA) (248 bp)[37], following a condition (30 cycled at 98 °C for 10 s, 58 °C for 30 s and 72 °C 10 s). DNA extracted from *C. diphtheriae* ATCC 13812 and *C. diphtheriae* ATCC 19409 which can produce diphtheria toxin was used as positive controls.

**Statistics**. Counts data from bacteria, viruses and fungi were imported into R (*v3.5.1*). Alpha diversity metrics (richness and diversity) and rarefaction were calculated using the phyloseq package (*v1.26.0*). Principal coordinate analysis (PCoA) based on Bray-Curtis dissimilarities of the microbial community structure were performed using the vegan package (*v2.5-3*). Heat maps were generated using the pheatmap package (*v1.0.10*) and was based on the Pearson correlation coefficient between different pathways in terms of their abundances across all samples. The proportion of taxa derived from donors in total taxa of patient's stool samples was calculated using fast expectation-maximization microbial source tracking (**FEAST**)[38].

**Reporting summary**. Further information on research design is available in the Nature Research Reporting Summary linked to this article.

## Data availability

16 S rRNA gene sequencing data are available in the NCBI Sequence Read Archive under BioProject PRJNA515137. Metagenomic sequence data generated from fungi-enriched fecal DNA and virus like particles (VLPs)-enriched fecal viral DNA for this study are available in the NCBI Sequence Read Archive under BioProject accession PRJNA641975.

The public reference databases used in this study are as follows: GRCh38 p12 (https://www.ncbi.nlm.nih.gov/assembly/GCF_000001405.38/), ChocoPhlAn database (http://huttenhower.sph.harvard.edu/humann2_data/chocophlan/chocophlan.tar.gz), UniRef90 universal protein reference database (https://www.uniprot.org/help/uniref), Metacyc database (https://metacyc.org/), Fungal database from HumanMycobiomeScan (https://sourceforge.net/projects/hmscan/), The complete NCBI viral RefSeq databases (https://www.ncbi.nlm.nih.gov/refseq/).

All other relevant data are available from the authors.

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

## Acknowledgements

We acknowledge a seed fund for Gut Microbiome Research provided by the Faculty of Medicine, The Chinese University of Hong Kong

## Author contributions

F.Z. and T.Z. performed the experiments. F.Z., T.Z. and Y.K.Y. conducted data analysis and drafted the manuscript. F.C., C.C.M. and C.K.L. performed allogenic haemotopoietic stem cell transplantation (allo-HSCT) and fecal microbiota transplantation (FMT) in clinic and provided clinical data. W.T., K.C. and M.H. collected the clinical samples and data. K.Y., Q.L. and C.P.C. helped with DNA sample preparation and data analysis. F.K. L.C. provided critical comments on the manuscript. S.N. and P.C. designed, supervised the study and revised the manuscript.

## Competing interests

The authors declare no competing interests.
