## [Peer Review File · Nature Communications]

Reviewers' comments:

Reviewer #1 (Remarks to the Author):

The authors have been responsive to the comments of the reviewers. However, the major concern remains: it is unclear how much can be learned from a descriptive analysis of one patient.

Reviewer #4 (Remarks to the Author):

The authors describe a longitudinal microbiome study of a GvHD patient serially treated with FMT. Although this is a single case report, this will be of broad interest to the field due to the transkingdom - bacteriome, mycobiome and virome approach and for furthering our understanding of their roles in FMT treatment. Authors found that successive FMT led to an increased bacteriome and mycobiome diversity and showed acquisition of donor microbes throughout the follow-up duration. However, the virome characterization is cursory.

1. The virome description is fairly superficial. Authors only describe the virome in terms of relative abundance of three groups of viruses: 1) TTV (Fig 5C), 2) Microviridae at the family level and 3) Caudovirales at the order level (Fig 5D).

Yet Fig 5C shows that there are other unaccounted eukaryotic viruses. Likewise, Fig 5D shows that there are other unaccounted prokaryotic viruses besides viruses from the Microviridae family and Caudovirales order. There are thousands of annotated bacteriophages classified within the Caudovirales order, so it is unconventional to only describe it at the Order level. Also, did the patient acquire viruses from the donor through FMT? This all is in contrast to the detailed profiling of bacteria (see Fig 2B, S3) and fungi (see Fig 4D, 4E, S5).

2. What was the selection criteria for the list of bacterial, fungal and viral species shown in Figure S9 and S10? This is a potentially important clarification because Fig S10 lists an unusually high number of enterobacteria phages for human stool. Part of the challenge is the lack of clarity about virus prevalence/abundance/etc for reviewer to evaluate (see point #1).

If these are the most dominant bacteriophages sequenced, the authors should rule out contamination. Did the authors have NGS controls for contamination (e.g. reagent contaminants)?

3. Statistical test (one sample t test) is inappropriate, because the samples are not independent. i.e. longitudinal of single individual

4. There are a number of inconsistencies between the figures and legends. This was very disruptive to the review process. To name a few:

a. Figure 5D and 5E are inconsistent with their figure legends.

b. Figure S6: text refers to virome, but figure legend refers to it as mycobiota. In good faith, reviewer assumes plot is on virome and evaluated it as such.

c. Figure S10: text refers to bacterial-viral correlation, but figure legend refers to it as bacterial-fungal.

d. There are 2 Supplementary Table 2 – the VLP table should be table 3?

Reviewer #5 (Remarks to the Author):

I commend the authors on putting together an interesting manuscript.

The authors did a thorough job addressing all of Reviewer #3's previous comments, particularly addressing the (now we see unlikely) possibility that *Corynebacterium* infection was the primary cause of diarrhea.

One major issue from my reading of the manuscript:

The fungal data looks unlike any other study.

The data in Fig 4b and e is remarkably diverse. Quite a few papers have come out now showing that the fungi in human stool samples is low diversity and taxonomically rather variable from sample to sample (for example, see Zuo et al. 2018). In fact, the research from some groups like Auchtung et al. (2018) suggest that all or nearly all the fungi found in the stool of healthy individuals is ultimately from saliva (e.g., *Candida*) or consumed food (e.g., *Saccharomyces*). Aside from the one spike in *C. albicans*, the taxa here are very different from most other studies. Also, Donor 4 and Donor 8's samples are remarkably (and unbelievably) similar.

Could there have been contamination somewhere in the pipeline? Did the authors do negative controls throughout?

Another possibility is an issue with the analysis. Although I have not used it recently, as of a couple years ago, the NCBI fungal database contained a fair bit of bacterial contamination. There are also some fungal regions that genuinely share sequence homology with bacteria and therefore could lead to false hits. Did the authors follow up on the fungal metagenomic results by either assembling larger segments of fungal genomes or by doing ITS analysis on some of the same samples?

I would not be surprised if *Fusarium oxysporum* and *Botrytis cinerea* were indeed abundant before the FMTs. However, I am skeptical that a healthy mycobiome displaced these fungi. I think it rather more likely that a healthy bacterial community (and recovered immune system) displaced these fungi.

Other issues:

1. Wording issues:

Line 319: fix "we can hypothesis"

Fix Fig. 2 Legend Line 24 "whereas an only single stool"

2. Did the authors find any archaea? The methods do not mention that archaea were filtered from 16S rRNA gene results, though it is a possibility that the primers were biased against archaea and/or the patient and donors contained no archaea (unlikely because archaea are found in nearly everyone). Although archaea presumably are not as clinically relevant as viruses and fungi, if the authors have the data it would be interesting to know what happened to the archaeal community.

3. In case this is not caught at the proofing stage, I want to point out that since the authors did not do reverse transcription, they amplified the 16S rRNA gene.

For example, see:

Line 58

Fig 2 and Fig. 2 Legend Lines 22/23 and 27

Also 16S not 16s (Figure 2)

4. I would recommend the authors avoid pre-extraction centrifugation of fungal samples in the future (Methods Line 55). Quite a few fungi, especially molds, do not centrifuge out and so could be selected against here.

5. Figure 1a does not appear to be referenced in the text.

6. Figure 1a is confusing in multiple respects.

-- Why are some things on the left side of Day 0 (bacterial analyses) and other things on the right side of Day 0 (fungal + viral analyses)?

-- What is "fecal enriched DNA extraction"?

-- The methods (line 70) suggest 16S rRNA genes were amplified from the DNA extracted via the fungi protocol, but this figure suggests otherwise.

-- What is the significance of 120? (at the top -- it is the only day not labeled -- perhaps add "final sample collected" or something along those lines)

7. Figure 4e:

I cannot match the colors with the taxa. I see the value of the stacked bar plot, but unless I'm misunderstanding something, the taxa names don't line up and so cannot hope to be matched to the colors in the figure. Perhaps just label the most abundant taxa? Then perhaps the full taxonomic data could be supplied in a supplemental Excel file.

8. Why are different methods and different figure construction techniques used in the different figures?

In particular:

-- 2b vs 4c (PCA vs PCoA)

-- 2a (Observed/Shannon/Faiths Phylogenetic Diversity) vs

3a (Shannon/Simpson) vs

4b (Shannon/Simpson) vs

5e (Chao 1/Shannon/Simpson)

Unless there is a good reason, this leads to unnecessary confusion. Also, there could be the perception that the data is being displayed differently in order to support a certain argument.

9. I would argue that the increased fungal-bacterial and viral-bacterial correlations over time (Lines 255-266) may not mean anything. If there is an increase in the number of bacterial taxa, there will be an increase in the number of correlations.

Point-to-Point response to reviewers' comments

Reviewer#1(Remarks to the Author):

The authors have been responsive to the comments of the reviewers. However, the major concern remains: it is unclear how much can be learned from a descriptive analysis of one patient.

Response: Thank you for the comments. Previous study has reported changes in the bacterial community after FMT, while analysis of this case is the first to show that multiple FMTs can lead to significant and distinct alterations in the diversity and composition of not only the bacterial component but also the fungal and viral components of the gut microbiome. The serial and comprehensive datasets with multiple time points are useful in highlighting multi-kingdom (bacteriome, mycobiome

and virome) alterations and new approach in furthering our understanding of their roles in FMT treatment. Although the data are from a single case, we believe these data are novel and of broad interest to the FMT field and can help stimulate further multi-kingdom microbial studies in FMT.

Reviewers #4 (Remarks to the Author):

The authors describe a longitudinal microbiome study of a GvHD patient serially treated with FMT. Although this is a single case report, this will be of broad interest to the field due to the transkingdom - bacteriome, mycobiome and virome approach and for furthering our understanding of their roles in FMT treatment. Authors found that successive FMT led to an increased bacteriome and mycobiome diversity and showed acquisition of donor microbes throughout the follow-up duration. However, the virome characterization is cursory.

Response: We thank the reviewers for the positive feedback. We have performed further comprehensive virome analysis and included the data as below.

1. The virome description is fairly superficial. Authors only describe the virome in terms of relative abundance of three groups of viruses: 1) TTV (Fig 5C), 2) Microviridae at the family level and 3) Caudovirales at the order level (Fig 5D). Yet Fig 5C shows that there are other unaccounted eukaryotic viruses. Likewise, Fig 5D shows that there are other unaccounted prokaryotic viruses besides viruses from the Microviridae family and Caudovirales order. There are thousands of annotated bacteriophages classified within the Caudovirales order, so it is unconventional to only describe it at the Order level. Also, did the patient acquire viruses from the donor through FMT? This all is in contrast to the detailed profiling of bacteria (see Fig 2B, S3) and fungi (see Fig 4D, 4E, S5).

Response: Thank you for the suggestion. We have provided detailed data of eukaryotic and prokaryotic viruses in the patient and donors. We reported alterations of the most dominant 50 eukaryotic viral species following FMT, and found that eukaryotic viruses other than TTV viruses were present in low abundance and did not display significant changes up to 3 months after FMT (**Supplementary Figure 10a**) (Lines 247-250, RESULTS). We have also summarized alterations of prokaryotic viral community in the patient after FMT at the species level (**Supplementary Figure 10b**).

Before FMT, the patient's prokaryotic viral community was dominated by *Parabacteroides phage YZ-2015a* (74.3%) and *Parabacteroides phage YZ-2015a* (13.9%), which decreased to 0% at 4 months after FMT. After the second FMT, five *Enterobacteria phage* species, one *Escherichia phage* species, two *Stx2 converting phage* species and one *Shigella virus* species expanded transiently, which were depleted after the fourth FMT. In contrast, five *Enterococcus phage* species maintained at high relative abundance from day 67 to 120 (**Supplementary Figure 10b**) (lines 260-268, RESULTS). We further investigated the proportion of recipient viral community that was derived from donors. Donor's viruses gradually took up greater proportions in recipient after each FMT (**Fig. 5a**). Less than 2% of the viral taxa was transferred from donor after the first and second FMT however after the third FMT, the proportion of donor-derived viral taxa increased up to 29% by 3 months (lines 224-230). The above additional data have been included in the revised manuscript.

2. What was the selection criteria for the list of bacterial, fungal and viral species shown in Figure S9 and S10? This is a potentially important clarification because Fig S10 lists an unusually high number of enterobacteria phages for human stool. Part of the challenge is the lack of clarity about virus prevalence/abundance/etc for reviewer to evaluate (see point #1). If these are the most dominant bacteriophages sequenced, the authors should rule out contamination. Did the authors have NGS controls for contamination (e.g. reagent contaminants)?

Response: Thank you for the comments. The bacterial, fungal and viral species were selected based on their relative abundance and we selected the top 40 most abundant bacterial, fungal and viral species. We have removed Figure S9 and S10 as suggested by a subsequent reviewer. We found that enterobacteria phages were the most dominant bacteriophages identified on sequencing. To preclude the possibility of contamination, we sequenced a negative control sample (reagent control) when we performed VLP extraction and amplification. DNA concentration of the negative control was 8.60 ng/ul which was substantially lower than the fecal sample VLP preparations (averaged DNA concentration 83.7 ± 24.3 ng/ul). Sequencing data showed that only five species were found in the negative controls; *Burkholderia virus BcepF1*, *Pseudomonas virus JG024*, *Escherichia virus RCS47*, *Pseudomonas virus tabernarius* and *BeAn 58058 virus* were detected (**Supplementary Figure 11**). These five species

collectively took up less than 0.4% of the viral communities across all donors and recipient samples. These data suggest that viral alterations reported are unlikely to be the result of contamination (Lines 268-274, RESULTS SECTION).

3. Statistical test (one sample t test) is inappropriate, because the samples are not independent. i.e. longitudinal of single individual

Response: Thank for the comments. We have removed the description of this test in the text and figures (**Fig. 5c and 5e**).

This was very disruptive to the review process. To name a few:

a. Figure 5D and 5E are inconsistent with their figure legends.

Response: We apologize for the inconsistencies and we have carefully amended the figure legends. **Fig. 5d** shows richness, shannon and simpson diversity of phages in donor and patient stools. **Fig. 5e** shows relative abundance of Microviridae and Caudovirales phages in donor and patient stools.

b. Figure S6: text refers to virome, but figure legend refers to it as mycobiota. In good faith, reviewer assumes plot is on virome and evaluated it as such.

Response: Thank you. Figure S6 refers to virome and the legend has been corrected in the revised manuscript as **Supplementary Figure 8b**.

c. Figure S10: text refers to bacterial-viral correlation, but figure legend refers to it as bacterial-fungal. .

Response: Thank you. We have removed **Fig 10S** as suggested by a subsequent reviewer.

d. There are 2 Supplementary Table 2 – the VLP table should be table 3?

Response: Thank you. The VLP table should be table 3 and this has been amended (Line 289, METHOD).

Reviewer #5 (Remarks to the Author):

I commend the authors on putting together an interesting manuscript.

The authors did a thorough job addressing all of Reviewer #3's previous comments,

particularly addressing the (now we see unlikely) possibility that *Corynebacterium* infection was the primary cause of diarrhea.

Response: We thank the reviewer for the positive comments on our paper.

One major issue from my reading of the manuscript: The fungal data looks unlike any other study. The data in Fig 4b and e is remarkably diverse. Quite a few papers have come out now showing that the fungi in human stool samples is low diversity and taxonomically rather variable from sample to sample (for example, see Zuo et al. 2018). In fact, the research from some groups like Auchtung et al. (2018) suggest that all or nearly all the fungi found in the stool of healthy individuals is ultimately from saliva (e.g., *Candida*) or consumed food (e.g., *Saccharomyces*). Aside from the one spike in *C. albicans*, the taxa here are very different from most other studies. Also, Donor 4 and Donor 8's samples are remarkably (and unbelievably) similar. Could there have been contamination somewhere in the pipeline? Did the authors do another possibility is an issue with the analysis. Although I have not used it recently, as of a couple years ago, the NCBI fungal database contained a fair bit of bacterial contamination. There are also some fungal regions that genuinely share sequence homology with bacteria and therefore could lead to false hits. Did the authors follow up on the fungal metagenomic results by either assembling larger segments of fungal genomes or by doing ITS analysis on some of the same samples?

Response: We are grateful to the reviewer for this important comment. We included a negative control group when we performed fecal fungal DNA extraction. The DNA concentration of the negative control was too low to perform metagenomic sequencing, suggesting that there is likely to be negligible fungal DNA contamination. We have also performed gut fungal profiling using ITS analysis. We analyzed the fungi profile by aligning sequence reads to fungal ITS database via *Bowtie 2.20*, in order to preclude bacterial sequence contamination. Moreover, we performed quantitative PCR on a subset of significantly altered fungal species. The consistency between fungal ITS profiling and qPCR results further confirms the validity of our findings. We have shown that fungal communities of donor 4 appeared to cluster away from Donor 8's samples (**Fig.4 b**) indicating their composition were different. The methods and results have been included in the manuscript in lines 187-215 (RESULTS) and 165-171, 185-203 (METHODS).

I would not be surprised if *Fusarium oxysporum* and *Botrytis cinerea* were indeed abundant before the FMTs. However, I am skeptical that a healthy mycobiome displaced these fungi. I think it rather more likely that a healthy bacterial community (and recovered immune system) displaced these fungi.

Response: We appreciate the reviewer's comment. We have included that after FMT a healthy bacterial community (and recovered immune system) likely displaced some of the over presented fungi present in the stool of the patient. We have included this in the manuscript in lines 325-327 (DISCUSSION).

Other issues:

1. Wording issues:

Line 319: fix "we can hypothesis"

Response: This has been removed in the discussion.

Fix Fig. 2 Legend Line 24 "whereas an only single stool"

Response: We amended to "whereas only one fecal sample collected from donor D4" (Line 24, FIGURE LEGEND).

2. Did the authors find any archaea? The methods do not mention that were filtered from 16S rRNA gene results, though it is a possibility that the primers were biased against archaea and/or the patient and donors contained no archaea (unlikely because archaea are found in nearly everyone). Although archaea presumably are not as clinically relevant as viruses and fungi, if the authors have the data it would be interesting to know what happened to the archaeal community.

Response: Thank you for your suggestion. We have performed archaea profiling using *Kraken2*. The method was shown in lines 173-177 (METHODS). We found the patient showed similar archaeal diversity and phylum-level composition with donor (**Supplementary Figure 12a and 12b**). After serial FMTs, there were no significant changes in the archaeal diversity or the composition at the phylum level (**Supplementary Figure 12b**). At the species level, *Methanopyrus sp. KOL6* predominated in both the donor and patient archaea communities. The relative abundance of *Archaeoglobus veneficus* and *Thermococcus gorgonarius* in patient at

Day 0 was 15.4% and 11.1% respectively, and both markedly decreased to <0.3% (day 120) after FMT. In contrast, *Archaeoglobus sulfaticallidus*, *Thermococcus pacificus*, *Thermococcus sp. EXT12c* which showed a lower relative abundance in patient before FMT showed an expansion after FMT (**Supplementary Figure 12c**). The finding has been added to the RESULTS (Lines 276-289).

3. In case this is not caught at the proofing stage, I want to point out that since the authors did not do reverse transcription, they amplified the 16S rRNA gene.

For example, see:

Line 58

Fig 2 and Fig. 2 Legend Lines 22/23 and 27

Also 16S not 16s (Figure 2)

Response: Thank you for your suggestion. We amended the “16S rRNA sequencing” to “16S rRNA gene sequencing” in the INTRODUCTION SECTION (line 61), Figure legend (lines 27) and amended 16s to 16S in **Fig. 1a**.

4. I would recommend the authors avoid pre-extraction centrifugation of fungal samples in the future (Methods Line 55). Quite a few fungi, especially molds, do not centrifuge out and so could be selected against here.

Response: Thank you for your kind comment and we agree with the suggested procedure.

5. Figure 1a does not appear to be referenced in the text.

Response: We amended the manuscript to reference Fig. 1a in the METHOD section (Line 23) and the RESULT section (Line 82).

6. Figure 1a is confusing in multiple respects.

-- Why are some things on the left side of Day 0 (bacterial analyses) and other things on the right side of Day 0 (fungal + viral analyses)?

Response: Thank you. To make the flow clearer, we have provided an updated **Fig. 1a** and provided details in the METHODS.

-- What is "fecal enriched DNA extraction"?

Response: We amended this to “fecal DNA extraction” in **Fig.1a**

-- The methods (line 70) suggest 16S rRNA genes were amplified from the DNA extracted via the fungi protocol, but this figure suggests otherwise.

Response: We apologise for the inconsistency. 16S rRNA genes were amplified upon the total fecal DNA extracted using the DNeasy Power Soil Kit according to manufacturer’s instructions. For mycobiome (fungome) profiling and in-depth bacterial microbiome profiling, we performed fungi-enriched DNA extraction followed by ultra-deep metagenomic sequencing for synergistic profilings of mycobiome, bacteriome and archaea. We amended the information in the METHOD section in lines 53-55 and in the **Fig. 1a**.

- What is the significance of 120? (at the top -- it is the only day not labeled -- perhaps add "final sample collected" or something along those lines)

Response: 120 is the final date of sample collection. We added the label “ final follow-up” to the **Fig. 1a**.

7. Figure 4e:

I cannot match the colors with the taxa. I see the value of the stacked bar plot, but unless I'm misunderstanding something, the taxa names don't line up and so cannot hope to be matched to the colors in the figure. Perhaps just label the most abundant taxa? Then perhaps the full taxonomic data could be supplied in a supplemental Excel file.

Response: We thank the reviewer for the kind suggestion. Fig.4e has been changed to Fig. 4d. All taxa names are lined up to match the colors in the figure. In addition to the figure, we have included the full taxonomic composition table as a supplemental file (**Supplementary dataset 1**).

8. Why are different methods and different figure construction techniques used in the different figures?

In particular:

-- 2b vs 4c (PCA vs PCoA)

-- 2a (Observed/Shannon/Faiths Phylogenetic Diversity) vs

3a (Shannon/Simpson) vs

4b (Shannon/Simpson) vs

5e (Chao 1/Shannon/Simpson)

Unless there is a good reason, this leads to unnecessary confusion. Also, there could be the perception that the data is being displayed differently in order to support a certain argument.

Response: Thank you for your kind suggestion. We have amended the diversity index of bacteria from Observed, Shannon, Faiths Phylogenetic matrix to Chao1, Shannon and Simpson matrix in **Fig. 2a**, and of the compositional interrogation from PCA to PCoA in **Fig. 2b**. The Chao1 index of genes and pathways were added in **Supplementary Figure 5**, and the Chao 1 index of fungi were supplemented in **Fig. 4a**

9. I would argue that the increased fungal-bacterial and viral-bacterial correlations over time (Lines 255-266) may not mean anything. If there is an increase in the number of bacterial taxa, there will be an increase in the number of correlations.

Response: Thank you. We agree with the reviewer. We have therefore removed Figure 9 and Supplementary 10 which showed fungal-bacterial and viral-bacterial correlations.

Reviewers' comments:

Reviewer #4 (Remarks to the Author):

The two main concerns from the prior review were:

- 1) virome characterization needed more characterization, and
- 2) concern about lack of contamination controls.

The authors have addressed both of these concerns with additional experiments and data. The new data shown in Supplementary Figure 10 is particularly striking. It might be worth moving it to primary figure as it conveys transmission dynamics (just a suggestion if space allows).

The authors included new analyses on the archaeal communities. That is outside my expertise. To my knowledge, it has its own challenges (limited database, etc.). So it would be good to have it looked at by an appropriate reviewer. In my opinion, that is a minor part of the manuscript that can be omitted if needed.

As mentioned in my prior comments, although this is a single case report, the longitudinal follow up and "multi-kingdom" analyses makes this of interest to the field.

Reviewer #5 (Remarks to the Author):

The authors did a very good job addressing my minor comments, but not my one major issue -- there is a critical problem with the WGS data or analysis of fungi, and possibly the new archaea data/analysis.

There have been numerous studies on fungi in the gut, and one constant between them is high variation -- both between different donors and between samples from the same donor. In this study, the fungal communities of two donors (and most from the FMT recipient) (Fig 4d) are extremely similar to each other. If this were true, it would be revolutionary to the field. However, the extensive controls required to support such a groundbreaking finding were not performed.

In addition, the fungal taxa that were identified are unlike what has been seen in previous studies. This suggests the fungal profiles are either due to contamination or flawed analysis of the data.

The archaea data was also remarkably similar across all examined subjects (Supp Fig 12c). There is little relevant literature on archaea in the gut with which to compare, but what exists (e.g., Raymann et al. 2017 and Koskinen et al. 2017) suggests there may be a problem with the finding of high similarity across samples and the unusual taxonomic composition of those samples (including a high proportion of hyperthermophiles).

Here are some more specific comments on the authors' response:

The DNA concentration of the negative control was too low to perform metagenomic sequencing, suggesting that there is likely to be negligible fungal DNA contamination.

I would still strongly recommend sequencing negative controls to identify low levels of fungi or fungal DNA that may be in, for example, saline, collection vessels, DNA extraction chemicals, or the metagenomic library preparation kit.

We have also performed gut fungal profiling using ITS analysis. We analyzed the fungi profile by aligning sequence reads to fungal ITS database via Bowtie 2.20, in order to preclude bacterial sequence contamination.

Why did the authors change their fungal analysis methods from the June draft (used only NCBI fungal RefSeq) to the September draft (used only the UNITE ITS database)? Was there a problem with the original approach? I see that the colors (and number of colors) are very different between the two versions, but I can't tell if the taxa have changed because as I previously mentioned, I don't understand how the colors in Fig 4e/nor 4d match up with the legend. If multiple taxa are going to share the same color, the order of the taxa in the legend is particularly important. Have the authors checked the accuracy of their pipeline(s) by analyzing a mock community? In this case, it may be appropriate to do multiple mock communities composed of different groups of organisms -- for example, are the right fungi identified from the fungal mock community? Are any fungi identified from the bacterial mock community? As mentioned previously, the authors could also confirm the validity of their pipeline by either 1) assembling larger segments of fungal genomes from the existing WGS data, or 2) directly PCR amplifying and sequencing ITS1 or ITS2.

Moreover, we performed quantitative PCR on a subset of significantly altered fungal species. The consistency between fungal ITS profiling and qPCR results further confirms the validity of our findings.

Based on the qPCR, I agree that these specific taxa are different between the single pre-FMT sample and most of the post-FMT and donor samples.

Although the qPCR evidence is limited (just 3 taxa), it does support the argument that most samples are similar to each other. If true, then the problem is likely contamination rather than incorrect analysis.

Also a word of caution about over-interpretation: I understand the difficulty in obtaining more pre-FMT samples, but since there was just an n=1 pre-FMT, it is hard to say whether this difference was due to chance or whether this snapshot was truly representative of the patient's mycobiome.

We have shown that fungal communities of donor 4 appeared to cluster away from Donor 8's samples (Fig.4 b) indicating their composition were different.

Ordination plots accentuate the differences between samples. If the samples are nearly identical, small differences will be magnified. If on the other hand, the donor samples were plotted together with some typical, diverse gut fungal samples, the donor dots would cluster even closer together.

Point-to-Point response to reviewers' comments

Reviewers' comments:

Reviewer #4 (Remarks to the Author):

The two main concerns from the prior review were:

- 1) virome characterization needed more characterization, and
- 2) concern about lack of contamination controls.

The authors have addressed both of these concerns with additional experiments and data. The new data shown in Supplementary Figure 10 is particularly striking. It might be worth moving it to primary figure as it conveys transmission dynamics (just a suggestion if space allows).

Response: We appreciate the reviewer's positive feedback. We have moved Supplementary Figure 10 to Primary Figure 5 (**Fig 5f** and **Fig 5g**).

The authors included new analyses on the archaeal communities. That is outside my expertise. To my knowledge, it has its own challenges (limited database, etc.). So it would be good to have it looked at by an appropriate reviewer. In my opinion, that is a minor part of the manuscript that can be omitted if needed.

Response: Thank you for the helpful comment. As suggested, we have removed analyses of the archaeal communities from the manuscript due to limited database and we have focused on the virome and fungome analyses.

As mentioned in my prior comments, although this is a single case report, the longitudinal follow up and "multi-kingdom" analyses makes this of interest to the field.

Response: Thank you very much for the positive comments.

Reviewer #5 (Remarks to the Author):

The authors did a very good job addressing my minor comments, but not my one major issue - there is a critical problem with the WGS data or analysis of fungi, and possibly the new archaea data/analysis.

There have been numerous studies on fungi in the gut, and one constant between them is high variation -- both between different donors and between samples from the same donor. In this study, the fungal communities of two donors (and most from the FMT recipient) (Fig 4d) are extremely similar to each other. If this were true, it would be revolutionary to the field. However, the extensive controls required to support such a groundbreaking finding were not performed. In addition, the fungal taxa that were identified are unlike what has been seen in previous studies. This suggests the fungal profiles are either due to contamination or flawed analysis of the data.

Response: We are grateful to the reviewer for this important comment. We have performed additional experiments including (i) sequencing negative controls to exclude the possibility of contamination, (ii) refining our pipeline/bioinformatics analysis using the most recent *HumanMycobiomeScan* (Soverini, M., et al. *HumanMycobiomeScan: a new bioinformatics tool for the characterization of the fungal fraction in metagenomic samples. BMC genomics* **20**, 496 (2019)) to confirm the accuracy of the WGS fungi analysis with the inclusion of additional controls, mock community validation, ITS2 sequencing and qPCR analysis, and (iii) removing the archaea analysis due to limited database. We have provided detailed response to each of the related specific comments below.

In line with data from several studies on the gut fungi, our new results showed variation of fungal communities between different donors and between serial samples from the same donor (**Fig 4**). In addition, fecal samples from 10 additional healthy controls were sequenced and compared to our donors' fecal mycobiome. These data showed that there is a large variation in the gut mycobiome among controls which are consistent with other reports in the literatures. The result is shown in **Response Figure 1**.

Response Figure 1. Species-level proportions of the fecal microbiota in additional healthy controls.

The archaea data was also remarkably similar across all examined subjects (Supp Fig 12c). There is little relevant literature on archaea in the gut with which to compare, but what exists (e.g., Raymann et al. 2017 and Koskinen et al. 2017) suggests there may be a problem with the finding of high similarity across samples and the unusual taxonomic composition of those samples (including a high proportion of hyperthermophiles).

Response: Thank you for the important comment. We agree with the current reviewer that there are limited literatures for archaea for comparison in existing database. As per this and the previous reviewer’s suggestion, we have removed the analyses of the archaeal communities from the manuscript due to limited database.

Here are some more specific comments on the authors' response:

The DNA concentration of the negative control was too low to perform metagenomic sequencing, suggesting that there is likely to be negligible fungal DNA contamination. I would still strongly recommend sequencing negative controls to identify low levels of fungi or fungal DNA that may be in, for example, saline, collection vessels, DNA extraction chemicals, or the metagenomic library preparation kit.

Response: Thank you for the great suggestion. We have included negative controls by using collection vessels and DNA extraction kit during DNA extraction. The amount of extracted DNA from the negative control was as low as 10 ng and 739,566 reads (0.2 G raw data) in contrast to a DNA amount of average 2 ug and a metagenomic output of 12 G from fecal samples. We found that the most abundant species in the negative controls were *Talaromyces pinophilus*, *Cryptococcus neoformans* and *Alternaria solani* but none of these species were detected in the recipient's or donors' samples. We have included these data in manuscript in results (lines 223-231) and **Supplementary Figure 7**. To further confirm these findings, we used quantitative PCR to measure *Parastagonospora nodorum*, *Thielavia terrestris*, *Candida dubliniensis* and *Saccharomyces cerevisiae* in stool samples of FMT recipient, donors and the negative controls. Similarly, these species were present in the samples of recipient and donors but were not detected in the negative controls. We have included these data in **Supplementary Figure 6**. Overall these results suggest that fungal alteration reported are unlikely to be the result of contamination.

Why did the authors change their fungal analysis methods from the June draft (used only NCBI fungal RefSeq) to the September draft (used only the UNITE ITS database)? Was there a problem with the original approach? I see that the colors (and number of colors) are very different between the two versions, but I can't tell if the taxa have changed because as I previously mentioned, I don't understand how the colors in Fig 4e/nor 4d match up with the legend. If multiple taxa are going to share the same color, the order of the taxa in the legend is particularly important.

Response: Thank you for the comments. In the June draft, we used the database NCBI fungal RefSeq data and we are very grateful to the reviewer for highlighting the potential limitation of the NCBI fungal database. Therefore, we attempted to improve our analysis methods by using the UNITE ITS database to eliminate bacterial contamination in a subsequent version (September draft) but we found that the pipeline failed to detect several species in mock communities. Since then a new pipeline *HumanMycobiomeScan*¹ published in June 2019 was reported to show an improved and enhanced performance to characterize fungal taxonomy from metagenomic data. The fungal community profile generated by this pipeline recapitulated our mock community composition and was also consistent with that of our newly conducted ITS2 sequencing result on a subset of our samples (these results can be

found below in the response to the next 2 concerns of the reviewer). We appreciate the reviewer's suggestion in helping us to improve our analysis by selecting the optimal pipeline. We have therefore reanalyzed all fungal composition by using *HumanMycobiomeScan*. These data have been included in the methods (lines 165-185), results (lines 188-221), abstract (lines 13-16), discussions (lines 305-309, and lines 320-328) and **Fig 4**. We have also updated the colour to be discriminatory and marked the key species in red color for clarity in **Fig 4d**.

Have the authors checked the accuracy of their pipeline(s) by analyzing a mock community? In this case, it may be appropriate to do multiple mock communities composed of different groups of organisms – for example, are the right fungi identified from the fungal mock community? Are any fungi identified from the bacterial mock community?

Response: Thank you for this important suggestion. We have checked the accuracy of our pipelines by analyzing 3 mock communities. We simulated three synthetic mock communities by *Grinder*, each of which was a human-associated metagenomic dataset consisting of 12 fungal, 5 bacterial and 1 viral species, as per the reviewer's suggestion. We used a dedicated fungi-centric pipeline *HumanMycobiomeScan* as it showed a good performance to characterize fungal taxa from metagenomic data, and we successfully detected all 12 fungal species with good prediction of fungal abundances in the 3 mock communities with only ~1% misassigned reads generated in each dataset. The result was shown as below (**Response figure 2**). Furthermore, *HumanMycobiomeScan* has the capability to filter bacterial and human reads using *BMTagger* to eliminate bacterial contamination during fungal profiling. The reference for this pipeline has been include in the manuscript (Soverini, M., et al. *HumanMycobiomeScan: a new bioinformatics tool for the characterization of the fungal fraction in metagenomic samples. BMC genomics* **20**, 496 (2019))

Response Figure 2: A comparison between the synthetic relative abundances and the predicted abundances via HumanMycobiomeScan (HMS) profiling of the constituent fungi in 3 mock communities. a. Mock community 1. b. Mock community 2. c. Mock community 3. The bars of HMS 1, 2, 3 indicate the reconstructed abundances via *HumanMycobiomeScan*. The bars of Mock community 1, 2, and 3 indicate the actual relative abundances. The other portion represents the fraction of misassigned reads.

As mentioned previously, the authors could also confirm the validity of their pipeline by either
 1) assembling larger segments of fungal genomes from the existing WGS data
 Or 2) directly PCR amplifying and sequencing ITS1 or ITS2

Response: Thank you for the suggestion. We have additionally performed both fungal ITS2 sequencing on a subset of stool samples and PCR validation on all stool samples. We found that ITS2 profiling (profiled via the pipeline PIPITS at the genus level) showed that *Candida* was the most predominant genus in the patient's fungal community, displaying approximately 99% relative abundance, which was consistent with the *HumanMycobiomeScan* profiling result on the metagenomic dataset (**Response figure 3**).

Response figure 3. A comparison between the profilings based on metagenomic sequencing and ITS2 sequencing analyses. a. Genus-level proportions of the fecal mycobiota in patient extracted from metagenomic sequencing and ITS2 sequencing. a. Genus-level proportions of the fecal mycobiota extracted for metagenomic sequencing data using *HumanMycobiomeScan*. b. Genus-level proportions of the fecal mycobiota extracted for ITS2 sequencing data by *PIPITS*. 25, 33, 35, 120 represented the stool samples that collected from patients at day 25, day 33, day 35, day 120. The ITS2 sequencing and analysis was performed according to Zuo, Tao, et al. "Gut fungal dysbiosis correlates with reduced efficacy of fecal microbiota transplantation in *Clostridium difficile* infection." *Nature communications* 9.1 (2018): 1-11.

In addition, we performed quantitative PCR to verify the findings obtained from *HumanMycobiomeScan*. We measured the significantly altered species, *Parastagonospora nodorum*, *Thielavia terrestris*, *Candida dubliniensis* and *Saccharomyces cerevisiae* in the FMT recipient and donors. Consistent with the sequencing results, *Parastagonospora nodorum* and *Thielavia terrestris* significantly decreased, *Candida dubliniensis* increased, whereas *Saccharomyces cerevisiae* transiently expanded in the feces of the recipient after serial FMTs. These data have been included in the methods (lines 196-215), results (lines 212-221), and **Supplementary Figure 6**.

Moreover, we performed quantitative PCR on a subset of significantly altered fungal species. The consistency between fungal ITS profiling and qPCR results further confirms the validity of our findings. Based on the qPCR, I agree that these specific taxa are different between the single pre-FMT sample and most of the post-FMT and donor samples. Although the qPCR

evidence is limited (just 3 taxa), it does support the argument that most samples are similar to each other. If true, then the problem is likely contamination rather than incorrect analysis.

Response: Thank you for the comments. As shown above, we have changed to a more updated and recent pipeline for fungi analysis, *HumanMycobiomeScan*, which provides more accurate analysis.

Also a word of caution about over-interpretation: I understand the difficulty in obtaining more pre-FMT samples, but since there was just an n=1 pre-FMT, it is hard to say whether this difference was due to chance or whether this snapshot was truly representative of the patient's mycobiome.

Response: Thank you for the suggestion. We have reworded the claims and included this as a limitation in the discussion (lines 352-354)

These are detailed as below:

Line 19 “FMT is effective in GvHD” was changed to “ FMT may be effective in GvHD

Line 142 “FMTs lead to large shifts” was changed to “FMTs was associated with large shifts”.

Line 181 “microbial community promptly recover” was changed to “microbial community might promptly recover”.

We have shown that fungal communities of donor 4 appeared to cluster away from Donor 8's samples (Fig.4 b) indicating their composition were different. Ordination plots accentuate the differences between samples. If the samples are nearly identical, small differences will be magnified. If on the other hand, the donor samples were plotted together with some typical, diverse gut fungal samples, the donor dots would cluster even closer together.

Response: Thank you for the comment. We completely agree with the reviewer. The new result obtained by *HumanMycobiomeScan* showed higher variation of fungal community both between different donors and between samples from the same donor. We have updated these results in **Fig 4**.

Reviewers' comments:

Reviewer #5 (Remarks to the Author):

I commend the authors on the significant amount of work done since the last revision!
Thank you for the additional controls and mock community analysis, and I really appreciate how you colored the Fig 4d key species in red.

Best of all was the use of the HMS program it seems, as the taxa are dramatically different than in the last version of the manuscript.

With this new analysis, your results look much more like I would expect with

- 1) the two donors no longer being virtually identical, and
- 2) the patient gaining typical host-associated fungi (Candida)

I now only have a couple minor suggestions:

Line 209: "colonized" and Line 325: "blooms"

Although the patient had compromised immunity and therefore nearly anything is possible, *Sporisorium reilianum* is a plant pathogen and it would be very unusual for a pathogen of plants to colonize an animal.

I think it is much more likely that this taxa is transiently present because the patient ate one of the common foods that this fungal species infects.

So I suggest avoiding terms that indicate this species is actively growing in the patient.

Also, this might have been caught in the proofing stage, but I happened to notice a typo in the legend for Supplementary Figure 4.

"umpped"

Point-to-Point response to reviewers' comments

Reviewers' comments:

Reviewer #5 (Remarks to the Author):

I commend the authors on the significant amount of work done since the last revision! Thank you for the additional controls and mock community analysis, and I really appreciate how you colored the Fig 4d key species in red.

Best of all was the use of the HMS program it seems, as the taxa are dramatically different than in the last version of the manuscript.

With this new analysis, your results look much more like I would expect with

- 1) the two donors no longer being virtually identical, and
- 2) the patient gaining typical host-associated fungi (Candida)

Response: We appreciate the reviewer's positive feedback.

I now only have a couple minor suggestions:

Line 209: "colonized" and Line 325: "blooms"

Although the patient had compromised immunity and therefore nearly anything is possible, *Sporisorium reilianum* is a plant pathogen and it would be very unusual for a pathogen of plants to colonize an animal.

I think it is much more likely that this taxa is transiently present because the patient ate one of the common foods that this fungal species infects.

So, I suggest avoiding terms that indicate this species is actively growing in the patient.

Response: Thank you for the suggestion. We have amended the statement.

Line 234-235: "*Sporisorium reilianum* derived from the donor colonized successfully in the patient's gut" was changed to *Sporisorium reilianum* derived from the donor showed high abundance in the patient's gut"

Line 351: “FMT was associated with blooms of several fungal species” was changed to “Several fungal species were increased after FMT”

Also, this might have been caught in the proofing stage, but I happened to notice a typo in the legend for Supplementary Figure 4.

"umpped"

Response: Thank you for the comment. We have amended to “unmapped” in the figure legend in Supplementary Figure 4.